

# *In silico* predictions of protein interactions between Zika virus and human host

João Luiz de Lemos Padilha Pitta[1,*], Crhisllane Rafaele dos Santos Vasconcelos[2,*], Gabriel da Luz Wallau[3], Túlio de Lima Campos[4] and Antonio Mauro Rezende[1]

[1] Microbiology Department, Aggeu Magalhães Institute-FIOCRUZ/PE, Recife, PE, Brasil
[2] Genetics Department, Federal University of Pernambuco, Recife, PE, Brasil
[3] Entomology Department, Aggeu Magalhães Institute-FIOCRUZ/PE, Recife, PE, Brasil
[4] Bioinformatics Platform, Aggeu Magalhães Institute-FIOCRUZ/PE, Recife, PE, Brasil
* These authors contributed equally to this work.

## ABSTRACT

**Background:** The ZIKA virus (ZIKV) belongs to the *Flaviviridae* family, was first isolated in the 1940s, and remained underreported until its global threat in 2016, where drastic consequences were reported as Guillan-Barre syndrome and microcephaly in newborns. Understanding molecular interactions of ZIKV proteins during the host infection is important to develop treatments and prophylactic measures; however, large-scale experimental approaches normally used to detect protein-protein interaction (PPI) are onerous and labor-intensive. On the other hand, computational methods may overcome these challenges and guide traditional approaches on one or few protein molecules. The prediction of PPIs can be used to study host-parasite interactions at the protein level and reveal key pathways that allow viral infection.

**Results:** Applying Random Forest and Support Vector Machine (SVM) algorithms, we performed predictions of PPI between two ZIKV strains and human proteomes. The consensus number of predictions of both algorithms was 17,223 pairs of proteins. Functional enrichment analyses were executed with the predicted networks to access the biological meanings of the protein interactions. Some pathways related to viral infection and neurological development were found for both ZIKV strains in the enrichment analysis, but the JAK-STAT pathway was observed only for strain PE243 when compared with the FSS13025 strain.

**Conclusions:** The consensus network of PPI predictions made by Random Forest and SVM algorithms allowed an enrichment analysis that corroborates many aspects of ZIKV infection. The enrichment results are mainly related to viral infection, neuronal development, and immune response, and presented differences among the two compared ZIKV strains. Strain PE243 presented more predicted interactions between proteins from the JAK-STAT signaling pathway, which could lead to a more inflammatory immune response when compared with the FSS13025 strain. These results show that the methodology employed in this study can potentially reveal new interactions between the ZIKV and human cells.

Corresponding authors
João Luiz de Lemos Padilha Pitta,
jlpitta82@gmail.com
Antonio Mauro Rezende,
antonio.rezende@fiocruz.br

## INTRODUCTION

Currently, there are approximately 80 species of arbovirus (arthropod-borne virus) that can infect humans. Arboviruses are a public health problem, mainly in tropical and developing countries, as seen by the annual epidemics of Dengue virus (DENV) as well as emergent arboviruses such as the Zika (ZIKV) and Chikungunya (CHIKV) viruses (*Bichaud et al., 2014*). ZIKV was highlighted in 2015 due to the high number of infections in Brazil and the correlation of ZIKV infection with increasing cases of Guillain–Barré syndrome and microcephaly in newborns (*Ayres, 2016*). Those facts led the World Health Organization to declare ZIKV a global health problem (*WHO, 2016*). Brazil had more than 200,000 reported cases of ZIKV at the peak of the 2016 emergency (*Spitz, 2019*). The number of cases has now decreased in Brazil, but ZIKV continues to circulate and there is still the possibility of further outbreaks.

ZIKV is a single-stranded positive-sense RNA virus, and it belongs to the genus *Flavivirus*, with a genome of 10.794 bp length encoding a polyprotein of approximately 3,400 amino acids (*Faye et al., 2014*; *Saiz et al., 2016*; *Sirohi et al., 2016*). In its mature phase, the polyprotein is cleaved into three structural and seven non-structural proteins (*Saiz et al., 2016*; *Sirohi et al., 2016*). From a phylogenetic point of view, ZIKV has two major lineages: one includes the African strains and the other is the Asian strains (*Saiz et al., 2016*). Many studies have investigated and found the association between the virus infection and microcephaly (*Hazin et al., 2016*; *Sirohi et al., 2016*; *Souza et al., 2016*; *de Araújo et al., 2017*), and also the sexual transmission, transmission between domestic animals and humans, and different mosquito vectors involved with the virus transmission (*Bueno et al., 2016*; *Aliota et al., 2017*; *Murray et al., 2017*; *Lowe et al., 2018*). Despite all the knowledge acquired about ZIKV and the confirmation that the infection causes microcephaly (*de Araújo et al., 2017*), there are still gaps about the biological processes related to ZIKV infection and spreading throughout the human body. A better understanding of the molecular mechanisms behind these processes will certainly provide key information to elaborate or improve strategies to combat this pathogen.

A way to increase the understanding of molecular events that take place between host and pathogens is the identification of protein interactions between them. However, the identification of protein interaction by experimental methods is a laborious process, costly, and subject to systematic errors, especially when applied on a large scale (*Jansen & Gerstein, 2004*; *Harrington, Jensen & Bork, 2008*). Due to these difficulties, computational methods emerge as important approaches once they are faster and cheaper. In addition, there is a daily increase in the amount of molecular information available in public databases, providing the raw information needed to generate more precise and sensitive computational models with greater capacity for generalization of methods and data representation. A sample of how fast new biological data has been deposited in different databases is presented by *Goodacre et al. (2020)*.

Computational analysis for PPI has been used in the most different data sources in recent years, for instance, *Shah et al. (2018)* and *Scaturro et al. (2018)* used a more traditional affinity purification-mass spectrometry (AP-MS) approach to generate data

and a computational methodology for build ZIKV PPI network. In *Shah et al. (2018)* flaviviruses were analyzed, including DENV and ZIKV, and conserved virus-host PPIs were compared to present evidence of the impacts on the immune system, while *Scaturro et al. (2018)* focused on the impacts of ZIKV infection on the neurological system. Other studies use a straight computational approach, collecting biological data from a public database and applying High-confidence Protein-Protein Interaction Prediction (HiPPIP) model (*Ganapathiraju, Karunakaran & Correa-Menéndez, 2016*) or protein interaction prediction techniques based on domain interactions (*Esteves et al., 2017*). *Ganapathiraju, Karunakaran & Correa-Menéndez (2016)* analyzed the functional and pathway associations of the interacting proteins, trying to find potential targets of ZIKV for host invasion and *Esteves et al. (2017)* employed an algorithm to predict PPI and compare targets and strategies of ZIKV and other *Flavivirus*. There are still studies that apply a computational approach to analyze a broad number of viruses, including ZIKV, as proposed by *Lasso et al. (2019)*.

Another useful set of approaches to predict PPI is the machine learning techniques. *Lian et al. (2019)* employed the Random Forest technique to predict PPI between bacteria and host, while *Ahmed, Witbooi & Christoffels (2018)* applied Support Vector Machine (SVM) to predict human-HPV PPI, and both used random sampling approach for building the negative dataset for the model.

Here, we used two machine learning methods (Random Forest and SVM) to predict protein interactions between two ZIKV strains and humans (*Homo sapiens*) applying the existing genomic and proteomic data for those organisms. A database of known validated protein interactions between different virus and hosts species was also used as a training set for the machine learning algorithms and the negative dataset used a dissimilarity-based negative sampling approach (*Eid, ElHefnawi & Heath, 2016*) to build the negative dataset, avoiding noise caused by random sampling technique (*Yang et al., 2020*). The features we used to train machine learning algorithms, as *Eid, ElHefnawi & Heath (2016)*, were based on the amino acid physicochemical properties of the proteins involved in the interactions of the training dataset, and then the models were applied in the prediction of interactions between the proteins of ZIKV and human. The *in silico* approach used here makes it possible to test the whole human proteome against the ZIKV proteins, which would demand a much longer time and would be overly expensive if performed using experimental methods. This study aims to provide clues on how the pathogenesis caused by ZIKV happens through PPI analysis using a computationally efficient yet robust solution.

## MATERIALS & METHODS

### Biological data

The Virus Pathogen database (https://www.viprbrc.org/) (*Pickett et al., 2012*) was used to obtain the ZIKV proteomes for the strains analyzed during the work. The strains PE243 (GenBank KX197192.1) and strain FSS13025 (GenBank MH158236) are both ZIKV strains of Asian lineages. The strain PE243 was isolated and sequenced in the state of Pernambuco–Brazil, during the Brazilian outbreak of 2015, and it was associated with

newborn microcephaly (*Donald et al., 2016*). This strain has been used in many studies (*Guedes et al., 2017*; *Chavali et al., 2017*; *Widman et al., 2017*; *Lima et al., 2019*), and it was selected due to the local epidemiological importance. FSS13025 is a strain isolated from Cambodia in 2010 and it was selected due to the evolutionary distance when compared to PE243 (*Widman et al., 2017*), as it is not associated with microcephaly cases and has been used for comparison with PE243 in other studies (*Aldo et al., 2016*; *Widman et al., 2017*; *Lima et al., 2019*). In addition, several experimental tools and data on these two ZIKV strains were obtained in our previous study, including expression data for infected human cells (*Lima et al., 2019*). Besides the ZIKV, the proteome of *Homo sapiens* (ID UP000005640) without isoforms was downloaded from the UniProt database (https://www.uniprot.org/) (*Bateman, 2019*) on February 7, 2019.

## Training interaction dataset

Classical machine learning approaches use positive and negative datasets to train classification algorithms. The present study uses a set of protein pairs interacting as a positive dataset. Those interactions come from the VirusMentha database (https://virusmentha.uniroma2.it/) (*Calderone, Licata & Cesareni, 2015*), which describes interactions between proteins of viruses belonging to 25 viral families and proteins from 8 different hosts. All interactions present in that repository have been obtained experimentally, and VirusMentha was chosen due to the diversity of host proteins that include bacteria, plants, metazoan, and proteins of 25 families of viruses. The diversity of information present in VirusMentha is desired for the machine learning approaches applied here to address the generalization and avoid biases in the trained model. The interaction information downloaded from VirusMentha is a tabular file with a confidence score for each protein interaction, however, it does not contain amino acid sequences for the proteins. The sequences of each protein present in the interactions were downloaded from the UniProt database. Interaction and protein sequence data were downloaded on February 7, 2019. Interactions in which one pair member was a viral polyprotein were removed due to the fact that viral polyproteins are cleaved to form viral mature proteins, and therefore have the amino acid sequence of all viral proteins, which could induce bias in the training data set. Interactions between two viral proteins were not used for the training dataset since the trained algorithms were used here to predict interactions between viral and host proteins. In addition, only interactions from VirusMentha with a confidence score higher than 0.3 were used, following the recommendations of *Villaveces et al. (2015)*.

The other step of dataset training is the negative dataset. The negative dataset is represented by pairs of proteins that probably do not interact. Selection of negative examples is a well-recognized challenge for PPI prediction since biological datasets rarely include pairs of proteins that are known not to interact (*Ben-Hur & Noble, 2006*; *Dyer, Murali & Sobral, 2011*), and there is no gold standard to generate non-interactions set (*Eid, ElHefnawi & Heath, 2016*; *Ahmed, Witbooi & Christoffels, 2018*; *Yang et al., 2019*). A common approach to generate non-interactions sets is randomly chosen protein pairs that are not present in the set of interacting proteins (*Ben-Hur & Noble, 2006*; *Eid, ElHefnawi & Heath, 2016*; *Ahmed, Witbooi & Christoffels, 2018*; *Mei & Zhang, 2019*;

*Yang et al., 2019*). However, previous studies have shown that random pairing may cause noise, limiting the usability of such negative sample sets (*Eid, ElHefnawi & Heath, 2016*; *Yang et al., 2020*). Since random sampling produces more incorrect negative examples than expected, degrading the learning process, and lowering the prediction sensitivity, our goal was to implement a process that finds pairs of proteins most likely to be non-interactive for machine learning use. In this work, the negative dataset was derived from the data downloaded from VirusMentha following the rationale of *Eid, ElHefnawi & Heath (2016)*, however, some adjustments were applied. First, we applied a local alignment comparing all proteins *vs.* all proteins present in the positive dataset using the Blastp program of the BLAST+ package version 2.2.30 (*Camacho et al., 2009*). Alignment parameters such as identity, similarity, and coverage were extracted, and only alignments with at least 80% of coverage between subject and query were used for the next steps. Clustering was also performed using CD-Hit version 4.6 (*Fu et al., 2012*) among the proteins from the positive dataset. A four-step approach was implemented for a viral protein "V" and host protein "H" to be considered a pair of proteins that do not interact. In the first step, the pair of proteins "V–H" must not be listed as a pair present in the positive dataset. The second and third steps are based on the methodology presented at *Eid, ElHefnawi & Heath (2016)*, where the second step consists in verifying if the protein "V" does not interact with another host protein that has at least 20% identity with "H". The third step verifies if the protein "H" does not interact with another viral protein that has at least 20% identity with "V". Additionally to what is proposed in Eid et al. (*Eid, ElHefnawi & Heath, 2016*), a fourth step verifies if another pair of a viral protein "A" and host proteins "B", where "A" has at least 40% of identity with "V" and "B" has at least 40% of identity with "H", is listed in the positive dataset. Therefore, if the pair "A–B" has at least 40% of identity with pair "V–H", the pair "V–H" is discarded alike a pair from the negative dataset due to the identity with pair "A-B" present in the positive dataset. Figure 1 shows the four steps approach.

The thresholds of steps two and three were set in 20% of identity as below that, the structural similarity is minimal (*Eid, ElHefnawi & Heath, 2016*). The 40% of identity used during the clusterization was set as it is the smaller threshold allowed by CD-HIT and the evaluation of identity between pairs aims to place in negative dataset pairs with the greatest possible difference of pairs listed in the positive dataset. The same number of positive pairs were generated following the steps above for the negative dataset, avoiding bias during algorithm training as proposed by *Shen et al. (2007)*.

## Algorithm training

Protein interaction predictions were performed using two classical techniques of machine learning: Support Vector Machine (SVM) and Random Forest. SVM and Random Forest were selected since both are well-validated algorithms for binary classification and have been used in many studies for the classification of biological data (*Statnikov & Aliferis, 2007*; *Statnikov, Wang & Aliferis, 2008*; *Gajowniczek et al., 2020*). We used the Random Forest algorithm as prediction tools implemented in Caret package version 6.0–78 for R (*Kuhn, 2013*), and SVM classification was done using LIBSVM version 3.22

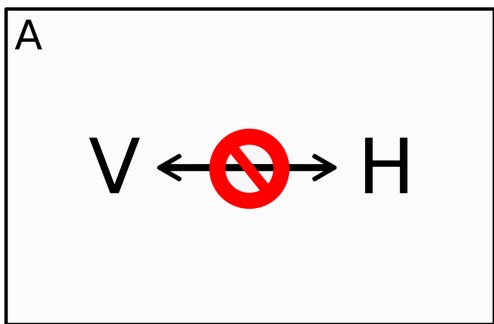
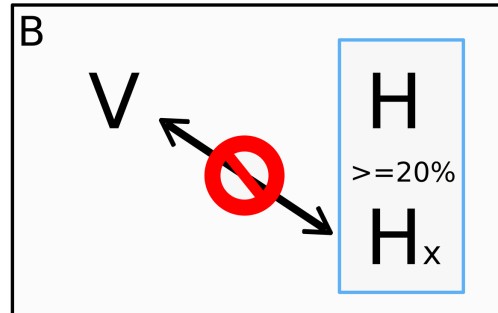
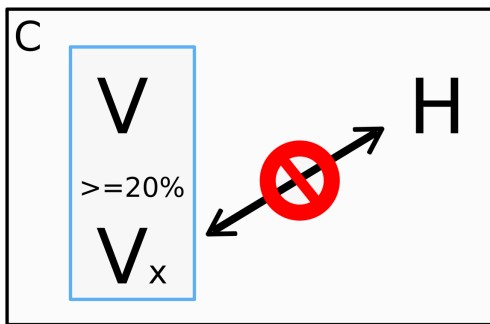
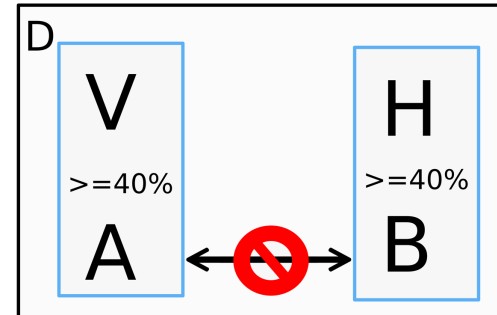

**Figure 1 Four steps approach for negative pair of proteins.** The four steps to set a pair of one viral protein "V" and one host's protein "H" as proteins that does not interact. The first step (A) verifies if the pair of proteins is not listed as a pair on the positive interaction's dataset. The second step (B) consists in verifying if the "V" protein does not interact with another host´s protein "Hx" that has more than 20% identity to "H". The third step (C) also checks if the protein "H" does not interact with another viral protein "Vx" that has more than 20% identity to "V". The fourth step (D) verifies if another pair, composed by a viral protein "A" that has at least 40% of identity with "V" and a host protein "B" that has at least 40% of identity with "H", is listed as positive interaction in the positive dataset. The pair "V–H" is discarded from negative dataset if it has at least 40% identity with pair "A–B" and "A–B" is listed in the positive dataset.

(*Hsu, Chang & Chin, 2008*) as the implementation of the SVM algorithm. The Random Forest algorithm was applied using two different modes named "class" and "prob". When mode "class" is used, the interaction prediction is given as label "0" for negative, or "1" for positive. Mode "prob" returns the probability of non-interaction associated with the prediction.

The features, based on physicochemical properties of proteins used to train both machine learning algorithms, were extracted using the R package protR version 1.4-2 (*Xiao et al., 2015*). Briefly, amino acid sequences of the proteins present in the training dataset were loaded into the R environment using the RStudio as IDE (Integrated Development Environment). The R package protR was first used to check if all the amino acid sequences had only the 20 valid amino acids and sequences that did not respect that were removed from further steps. Afterward, amino acids in the protein sequences were classified into seven groups, based on the physicochemical characteristics (*e.g.*, dipoles and side-chain volume), and each group received a label as proposed by *Cui, Fang & Han (2012)*. Table 1 shows the groups of amino acids and respective labels.

The protein sequences of amino acids became sequences of group labels, following the categorization scheme present in Table 1. After that, each sequence was analyzed in a triad of labels, and the frequency of each triad is calculated as proposed by *Shen et al. (2007)*, applying a normalization of the frequencies in values between 0 and 1. Thus, the sequences were converted into a vector of 343 positions (seven groups arranged in triads, so $7^3 = 343$). Therefore, the generated vectors had the same size allowing comparison between proteins with different sequence lengths. The output of protR with the normalized vectors was used as features for Random Forest and SVM algorithms training. The CSV (Comma-separated values) files with the vectors of features used to training both algorithms can be found in the Supplemental Information (see Dataset S1 for the negative dataset in Random Forest, Dataset S2 for the positive dataset in Random Forest, and Dataset S3 for negative and positive datasets in SVM).

During the training steps, the dataset was divided into two parts, one with 75% of the dataset used as the training set, and the other with 25% was used as the test set for validation of the model. The proportions of positive and negative samples of interactions were kept for both parts. Training steps were performed using 5-fold cross-validation for both algorithms, shuffling the interactions present in both sets former cited. A radial kernel function was applied for the SVM algorithm, which required a definition of gamma and sigma values in the model construction. Thus, a grid search for parameter selection was performed using grip.py (*Hsu, Chang & Chin, 2008*) script of LIBSVM aiming to determine the best values of sigma and gamma based on the training dataset. The results of classification for the dataset used as test set during the 5-fold cross-validation was used to calculate the accuracy, sensitivity, and specificity of the model. The model built for both algorithms, including models for modes "class" and "prob" of Random Forest algorithm, is available in the Supplemental Information (see Model S1 for SVM, Model S2 for Random Forest mode "class" and Model S3 for Random Forest mode "prob").

## Prediction of protein interactions

The ZIKV strains and human (*H. sapiens*) proteins were processed following the same steps for proteins present in the training dataset. Briefly, the protein sequences were loaded and protR was applied again to verify if all proteins had only valid amino acids. Sequences were converted into vectors representing the physicochemical features. Those vectors were combined in a pair-wise manner being one member belonging to *H. sapiens* and the other member to one of the two ZIKV strains. The constructed models for Random Forest and SVM were used to classify 586,572 pairs of proteins, being 28 proteins from both ZIKV strains against all 20,949 proteins from the human proteome.

## Analysis of data enrichment

A consensus prediction was constructed based on the predictions made by both algorithms, The consensus consisted of all positive predictions in Random Forest using mode "class", positive in SVM, and smaller than 0.5 in Random Forest using mode "prob". The human proteins present at the consensus predicted interactions for both ZIKV strains were submitted to perform functional enrichment analysis on the DAVID webservice

**Table 1 Classification of amino acids in group labels.**

| Amino acid | Group label |
| --- | --- |
| {A,V,G} | 1 |
| {I,L,F,P} | 2 |
| {Y,M,T,S} | 3 |
| {H,N,Q,W} | 4 |
| {R,K} | 5 |
| {D,E} | 6 |
| {C} | 7 |

version 6.8 (https://david.ncifcrf.gov/home.jsp) (*Jiao et al., 2012*). The enrichment analysis performed here is a technique to verify if the frequency of a KEGG (Kyoto Encyclopedia of Genes and Genomes–(https://www.genome.jp/kegg)) pathway describer term was higher than expected in the list of human proteins predicted to interact with ZIKV proteins, indicating a biological involvement of such pathways in the ZIKV infection events. Therefore, the analysis was performed following the rationale that a set of metabolic pathways could be affected by the interactions of proteins with ZIKV proteins and this fact could provide evidence and clues on how the virus is behaving inside the human host. The complete list of enriched pathways with their *p-value* and FDR (False Discovery Rate), including non-statistically relevant terms for both strains, is available in the Table S1.

We opted to use KEGG (*Ogata et al., 1999*; *Kanehisa, 2019*; *Kanehisa et al., 2019*) terms to describe the enrichment analysis results since KEGG is a knowledge base for systematic analysis of gene functions, linking genomic information with higher-order functional information (*Ogata et al., 1999*), and it is a reference for biological interpretation of genome sequences and other high-throughput data (*Kanehisa et al., 2019*). Besides genomic and functional information, KEGG provides terms that are straightforward for biological interpretation. Only terms with a *p-value* smaller than 0.05, adjusted by FDR of 5%, were considered relevant for this study. In the Supplemental Information is possible to find the set of pathway images from KEGG's database with enriched statistically relevant data (see Fig. S1 for strain PE243 and Fig. S2 for strain FSS130025).

### PPI network

The consensus predictions were used to build a PPI network applying the Cytoscape 3.6 tool (*Shannon et al., 2003*). The topology of the network was built in such a way that ZIKV proteins of both strains analyzed were merged in the network. The human proteins were divided into proteins that interact with only strain PE243, proteins that interact only with strain FSS13025, and proteins that interact with both strains. The complete PPI network is available on Fig. S3.

## RESULTS

The file downloaded from VirusMentha had 11,776 interactions described. Once the polyproteins and interactions between any two viral proteins were removed, a total of

4,157 interactions were kept. During the ProtR pattern checking step, a viral protein and seven host proteins were discarded, and 4,150 interactions were left. Since the interaction score of confidence was set as 0.3, a total of 348 interactions were used as a positive set during algorithm training. The negative set was derived based on the VirusMentha database (11,776 interactions), and 1,040,816 negative interactions were produced using the methodology applied here. Aiming to keep the balance between the number of pairs in both positive and negative training sets, 348 pairs of negative interactions were randomly selected to be used in algorithm training. Merging positive and negative for the training of algorithms, a total of 696 pairs with a higher level of confidence in the interaction were obtained.

Based on 5-fold cross-validation approach, the specificity, sensitivity, and accuracy of the Random Forest approach were calculated being equal to 82.27%, 74.73%, and 78.16%, respectively. The grid search for parameter optimization to SVM obtained a gamma of 0.125, sigma 2.0, and an accuracy of 77,29%.

Taking the consensus prediction between the two Random Forest modes and SVM, a total of 17,223 pairs of proteins were predicted as positive interactions, being 7,932 for ZIKV strain FSS13025 and 9,291 for PE243. Figure 2 shows the number of consensus predictions of ZIKV strain and *H. sapiens*.

The complete table of PPI present on Fig. 2 is available on Table S2. The interactions were grouped to visualize the number of human proteins that interact specifically with one or both ZIKV strains as Fig. 3 shows. It is possible to see a greater number of specific interactions (1,384) for the PE243 ZIKV strain. The main reason for that is the number of interactions for pr protein when both strains are compared.

The human proteins present in the final interaction predictions were used as input to functional enrichment analysis on DAVID Bioinformatics Resources 6.8 (https://david.ncifcrf.gov) (*Jiao et al., 2012*). The complete table with all the enrichment results, including non-statistically significant ones, can be accessed on Table S1. The enrichment analysis returned 103 and 101 terms for FSS13025 and PE243 ZIKV strains, respectively. Pathways associated with neural development as PI3K-Akt signaling, Ras signaling, NF-kappa B signaling, and Axon guidance were found enriched during the analysis. Terms associated with viral infection and viruses related to TORCH syndrome such as Herpes simplex infection were also enriched. It was possible to recover terms associated with the ZIKV infection such as endocytosis and apoptosis process. Interestingly, PE243 presented a term related to the JAK-STAT signaling pathway process while FSS13025 did not. Figure 4 shows the enriched pathways with statistical relevance found and the different pathways for each strain.

## DISCUSSION

The present study applied an approach easy to deploy using classical machine learning algorithms yet robust and fast for large-scale analysis. Aiming to build a better training set, interactions with the virus polyprotein and interactions between any two virus proteins were removed from the training set to avoid biases in the algorithm training phase, due to the fact that polyproteins have the sequence of all proteins and the present study is
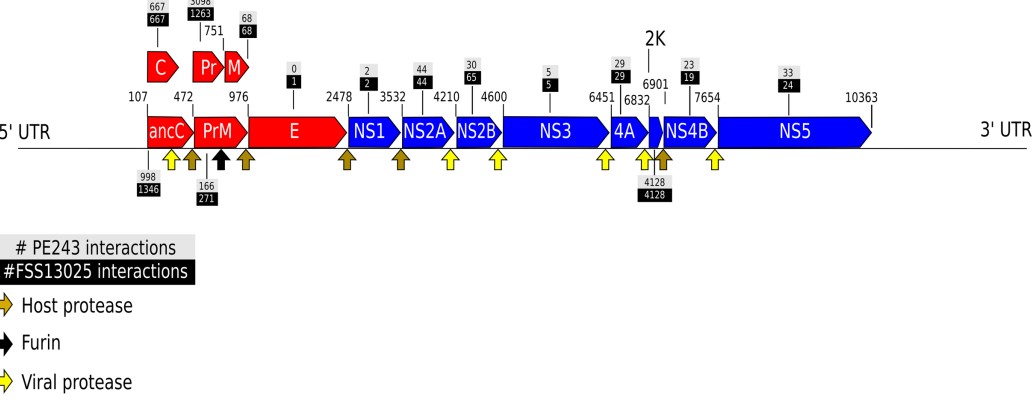

**Figure 2 Number of interactions predicted.** The number of interactions predicted by the consensus prediction was analyzed and was described, distributing the interactions by strain. Red and blue rectangles are structural and nonstructural ZIKV proteins, respectively.

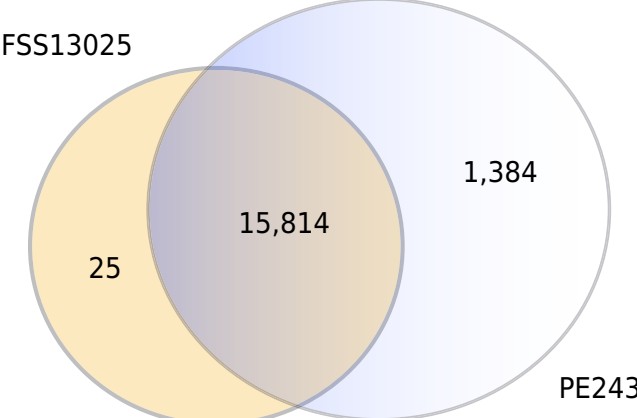

**Figure 3 Numbers of interactions per strain.** The Venn diagram shows the interactions of human proteins per ZIKV strain. The model predicted 25 exclusive interactions for strain FSS13025 while 1,384 exclusive interactions were predicted for strain PE243. The model predicted yet 15,814 interactions for both strains.

virus-host protein interaction classification. In addition, interactions with a confidence score value below the optimal value of 0.3 were discarded according to the cut-off point reported in *Villaveces et al. (2015)*, leaving a final set of 348 positive interactions.

In regard to the negative training set, we used the rationale proposed by *Eid, ElHefnawi & Heath (2016)*, instead of the common random sampling approach, to build a negative set with fewer false-negative samples as possible. This approach has been used in other studies like in *Yang et al. (2020)* and it implements a more accurate heuristic approach to generate non-interactive pairs. The methodology based on the study of *Eid, ElHefnawi & Heath (2016)* was applied, but unlike it, local alignment was used rather than a global strategy. The reason behind that is because specific protein domains may be important for the interactions (*Singhal & Resat, 2007*). Moreover, an additional step was implemented, comparing our approach to the one of *Eid, ElHefnawi & Heath (2016)*, to improve the

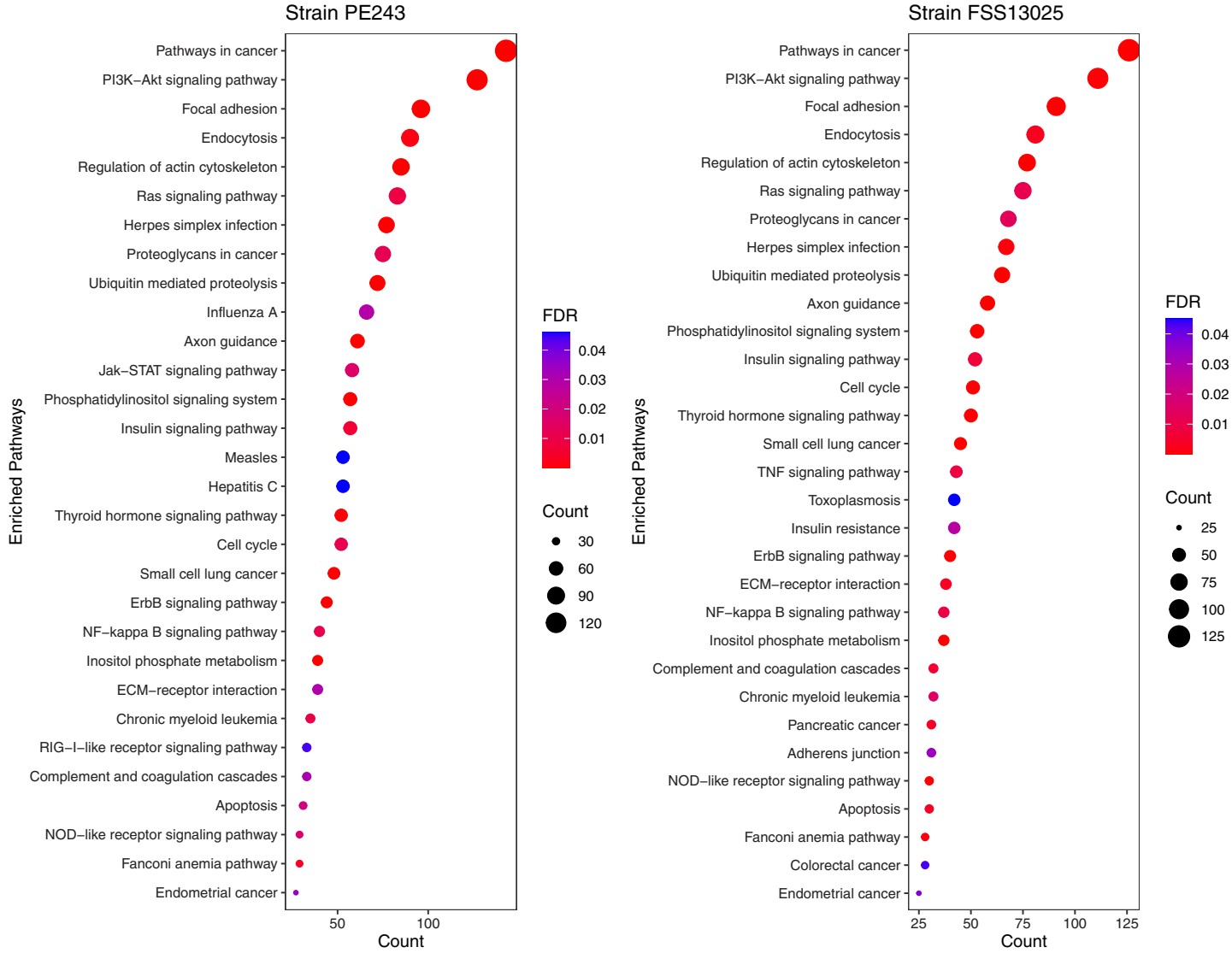

**Figure 4 Enriched pathways with statistical relevance.** The figure presents a dot plot graphic for each strain. The size of each dot represents the number of genes ("Count") from the consensus prediction enriched in each pathway and the color presents the statistical relevance based on the FDR.

confidence on negative interaction pairs. Therefore, to increase reliability, the sequence identity of possible pairs in the negative dataset was compared with pairs from the positive dataset to avoid not only identical but also similar positive interaction pairs. In the end, we had 1,040,816 negative pairs as a set of negative samples, and we selected 348 for the training of the algorithm to maintain the generalization of the model. The number of samples from the negative set was set equal to that of the positive set to avoid bias in the construction of the model as proposed in *Shen et al. (2007)*.

The process of extraction of physicochemical characteristics and normalization is similar to that used in other studies (*Dyer, Murali & Sobral, 2007*; *Cui, Fang & Han, 2012*), using the physicochemical characteristics extraction model proposed in *Shen et al. (2007)*. This model is the state of the art for this purpose in the type of data used here.

Proteins with non-standard amino acids were discarded, and the others were properly normalized using the R-implementation described in *Xiao et al. (2015)*. The proteomes of two ZIKV strains were compared to the *H. sapiens* proteome, but since physicochemical information of proteins is the core information, this approach could be used with any ZIKV strain, any virus, or any organism proteome that makes sense in biological terms.

The accuracy of the model was compatible with that of the methodology used in *Eid, ElHefnawi & Heath (2016)*, when using 75% of the interaction data in the training set and 5-fold cross-validation, and consequently comparable with other works for prediction between virus and host or prediction intraspecies. The results also showed that the increase in the number of cross-validation times for the dataset analyzed in this study did not improve significantly the accuracy of the model, contrary to what happens in other studies such as the one that applied cross-validation in the maximum likelihood of crystallographic simulated annealing refinement data (*Adams et al., 1997*).

The results regarding the number of interactions predicted by the model as positive are comparable to other studies that use as methodology Bayesian inference and protein interaction prediction techniques based on domain interactions (*Esteves et al., 2017*). *Yoon et al. (2017)* found 143 protein interactions between ZIKV and *H. sapiens in vitro*. Since other studies, including the present one, predict thousands of interactions *in silico*, there are strong indications that there is still much to be discovered about the molecular processes of infection by the virus.

Analyzing the number of interactions predicted by the model per ZIKV protein, it was possible to observe that the envelope protein of the strain FSS13025 was predicted with only one interaction and no interaction with the strain PE243. It is known that the envelope protein interacts with human proteins, thus we further investigated the prediction data. We observed a positive prediction between an envelope protein and protein ID P08709 (Coagulation factor VII) for the strain FSS13025. On the other hand, the strain PE243 got a score of 0.5 using Random Forest "prob" mode, which is exactly the threshold value for the same protein ID P08709. In addition, that interaction had a negative prediction for Random Forest "class" mode and a positive prediction for the SVM tool. Those facts show how difficult it is to predict interactions at the borderline score scale of the prediction tools. Therefore, it should be considered the fact that the model has an accuracy of around 78% when the two algorithms are merged in the consensus prediction. In parallel, that prediction found might be important since coagulation disorders in ZIKV infection have been reported as responsible for abnormal fetal growth due to reduced perfusion of the umbilical cord (*Anfasa et al., 2019*), and delayed fetal brain maturation (*Scher, 2019*).

Observing the Fig. 3 is possible to notice that strain PE243 showed a much higher number of exclusive interactions when compared with the exclusive interactions for the FSS13025 strain, which might corroborate that the PE243 strain is more virulent and associated with disorders like microcephaly when compared with the strain FSS13025 (*Lima et al., 2019*). A larger number of PE243's exclusive interactions happen with one ZIKV protein and, observing Fig. 2, pr protein is the one that has this larger number of interactions in strain PE243 when compared with FSS13025. Protein prM is associated

with ZIKV virulence (*Nambala & Su, 2018*) and is described for flaviviruses that after cleavage of prM in pr and M proteins, pr remains attached to the virion until exocytosis, preventing premature fusion in the trans-Golgi network (*Yu et al., 2009*; *Zheng, Umashankar & Kielian, 2010*), therefore helping the exportation of fully functional viral particles. Another interesting observation about the number of interactions of pr protein is that pr is a small protein when compared with others ZIKV proteins, and it was expected that a bigger protein, and consequently with a larger surface, would have more interactions. Size, however, is not the sole determinant and there are other examples in the literature describing, for instance, that hemoglobin exhibits greater surface activity than the much larger fibrinogen protein (*Dee, Puleo & Bizios, 2002*).

Since metabolic pathways can be affected by the interactions between ZIKV and human proteins, enrichment analysis, using the list of proteins ID of human proteome predicted as interacting with ZIKV proteins, was performed aiming to provide evidence and clues on how the virus is behaving inside the human host. Several pathways were identified as being affected by the interaction of ZIKV and human proteome. Some of them were found just for one out of two strains used in this work and others were recovered for both strains. Regarding the pathways found for both viruses, we noticed the presence of important pathways already described as being involved in virus infection such as the Inositol phosphate metabolism. This pathway has already been described as an important pathway for rabies virus (RABV), which causes acute encephalitis. *Besson et al. (2019)* showed phosphatidylinositol metabolism as the prominent factor for RABV infection, and those findings were confirmed in human neurons. Another interesting pathway that had several proteins involved (enriched) is the Fanconi anemia. More than half of proteins that participate in this pathway were predicted to interact with ZIKV proteins. This pathway is related to microcephaly and mental retard (*Krakow, 2013*). The ErbB signaling pathway is a clear enriched pathway for ZIKV since several viruses were described using this family of receptors to get into the cell. In addition, some viruses also can use this pathway to alter the cell cycle changing the speed at which they multiply (*Ho et al., 2017*). Moreover, we could see proteins from this pathway related to Glioma differentiation interacting with ZIKV proteins as shown in Fig. 5.

Finally, this pathway is involved in the activation of one of the main intracellular signaling pathways of the human metabolome such as the NF-kappa B signaling pathway, PI3K-Akt signaling pathway, and Phosphatidylinositol signaling system, all of them being found enriched for our data. Interestingly, another central pathway called the Ras signaling pathway was also enriched. It has already been described as an important pathway for some viruses such as *Herpesvirus*, and it is involved with several important biological processes such as differentiation, proliferation, and cell survival (*Filippakis, Spandidos & Sourvinos, 2010*). In our prediction, most proteins present in this pathway were predicted interacting with ZIKV proteins. We could see some expected enriched pathways normally modulated in viral infections such as apoptosis, cell cycle, and endocytosis, however, some more peculiar pathways were also identified such as the axon guidance pathway. The main cell controlling the axonal guidance is the astrocyte, and it is

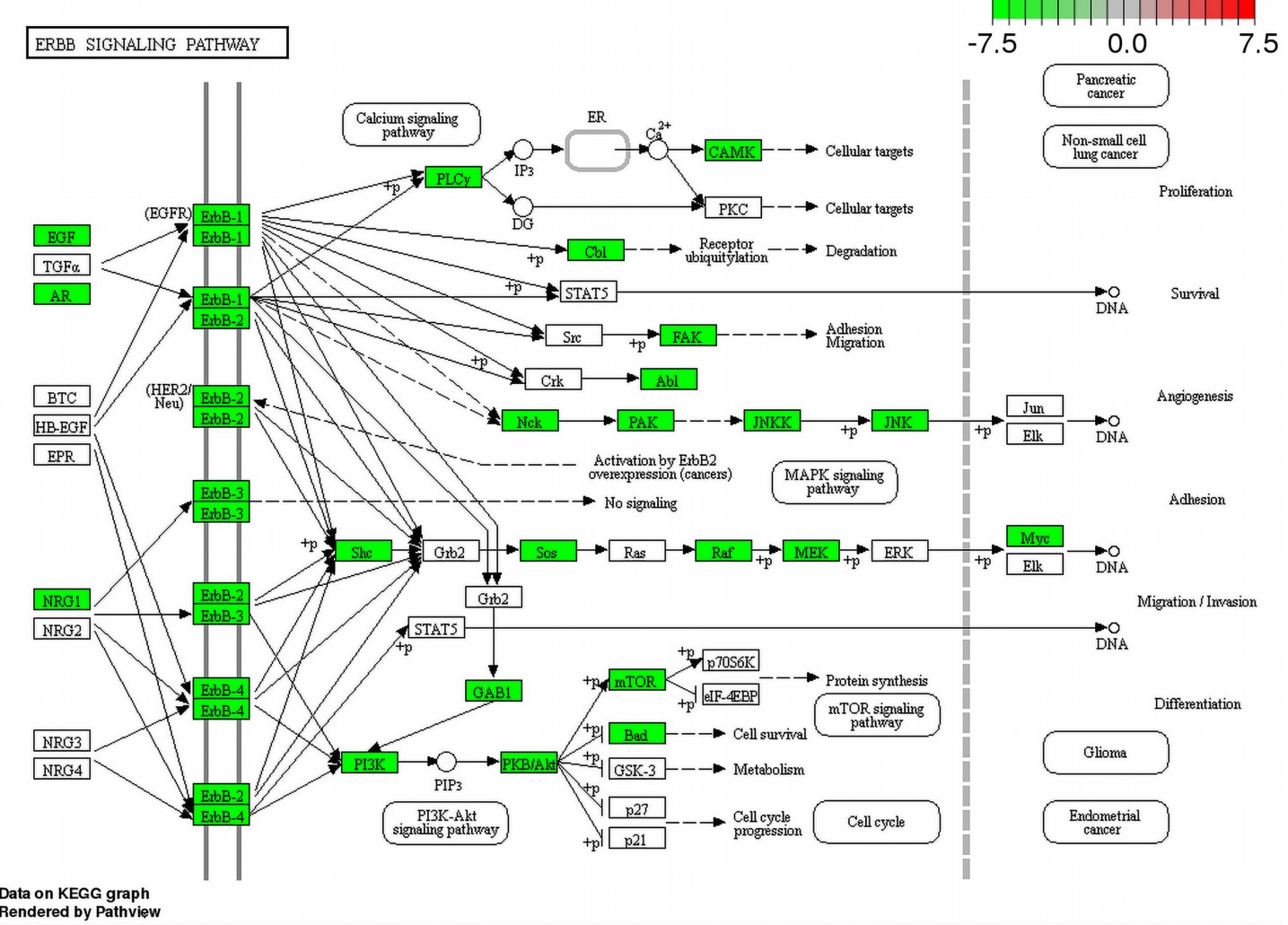

**Figure 5 ErbB signaling pathway map.** Enriched proteins in our data were marked in green over the pathway map. Green boxes represent proteins that were enriched by DAVID analysis between ZIKV (PE243 strain) and *H. sapiens* interactions.

one of the targets of ZIKV infection (*Limonta et al., 2018*; *Sher, Glover & Coombs, 2019*). This could be involved directly in microcephaly symptoms, and we could see several proteins from the host interacting with ZIKV proteins (*Limonta et al., 2018*).

Another interesting pathway that deserves a highlight is the complement and coagulation cascades. Some symptoms such as the rash on the skin (macules and papules) can be related to the interference in this pathway. It is worth mentioning ZIKV and Dengue virus belong to the same family, and those pathways are very disturbed on Dengue virus infection as it causes dengue hemorrhagic fever (*Rigau-Pérez et al., 1998*).

The extracellular matrix (ECM) interaction pathway along with focal adhesion structures are important sets of ZIKV protein interactions once the contact among cells (infected and uninfected) is important to viral spreading (*Mothes et al., 2010*). Viruses such as measles also can cause an acute inflammatory process in neurons, the contact between cells is important to guarantee the virus spreading (*Lawrence et al., 2000*). Moreover, it is

possible to notice using the KEGG representation of ECM and focal adhesion pathway how they are connected and how pathways already mentioned here, which are disturbed by ZIKV infection, are influenced. One example is the phosphatidylinositol signaling system, showing how the virus infection has a considerable impact on normal cell functioning.

Another impacted pathway in which many proteins are interacting with ZIKV proteins is ubiquitin-mediated proteolysis. Several viruses are known to manipulate the proteasome pathways aiming to change the cell cycle, inhibit apoptosis, evade the immune system, and activate cell signaling, helping the persistence of viral infection and viral carcinogenesis (*Shoji, 2012*).

Some cancer pathways were highlighted by enrichment analysis. One possibility is because all cancer pathways which appeared here, include other pathways already mentioned. This is clear in the KEGG pathway called "Pathways in Cancer" where all pathways represented in this scheme are interacting with proteins of ZIKV.

It is known that viral infection can affect insulin signaling pathways as it is modulated by the production of interferon-γ, a common molecule present in viral infection. However, the ZIKV was able to induce an insulin resistance pathway which can be a pathway activated as an immune response process (*Šestan et al., 2018*). In addition, this can be related to pancreatic cancer pathways, as the virus seems to induce an increase in insulin production.

On the other side, we had few pathways that were exclusively enriched for one of the strains of ZIKV studied here. The main one is the JAK-STAT signaling pathway. The JAK-STAT transducers and activators of transcription (JAK/STAT) pathway is one of a handful of pleiotropic cascades used to transduce a multitude of signals for development and homeostasis in animals, from humans to flies, and is considered the principal signaling mechanism for a wide array of cytokines and growth factors in mammals (*Rawlings, Rosler & Harrison, 2004*). Once the cognate receptor of STAT is bound to cytokines, it is activated by members of the JAK family of tyrosine kinases and, after activated, they dimerize and translocate to the nucleus and modulate the expression of target genes (*Kisseleva et al., 2002*). In addition to the activation of STATs, JAKs mediate the recruitment of other molecules such as the MAP kinases, PI3 kinase, etc. These molecules process downstream signals *via* the Ras-Raf-MAP kinase and PI3 kinase pathways which results in the activation of additional transcription factors (*Kisseleva et al., 2002*). We could see more proteins from the JAK-STAT signaling pathway interacting with proteins of ZIKV strain PE243 compared to the protein of strain the FSS13025, which could lead to a more disturbed immune response in the first one than in the second one. Interferons (IFNs) are the main group of proteins responsible for an innate immune response during viral infection and JAK/STAT is the main pathway activated by the IFNs (*Nan, Wu & Zhang, 2017*). A previous work of our group (*Lima et al., 2019*) shows that the PE243 strain can induce a higher expression of IFN-α, probably modulating the response of the JAK/STAT pathway stronger than the modulation performed by the FSS13024 strain.

At the end, it is worth mentioning our results are corroborated at some level by other works with different methodologies, including a previous work of our group (*Lima et al.,*

2019). A clear example appears in Scaturro et al. (*Scaturro et al., 2018*), since they found the modulation of proteins such as protein kinase B, also known as AKT, and ATM-ATR complex is regulated by ZIKV infection, and those proteins are also present in our results as interacting with ZIKV proteins. Another example is found in Shah et al. (*Shah et al., 2018*) where pathways such as regulation of apoptosis and ubiquitin-mediate protein degradation were found just like in our results. Scaturro et al. (*Scaturro et al., 2018*) and Shah et al. (*Shah et al., 2018*) used a more traditional approach with affinity purification coupled with liquid chromatography and tandem mass spectrometry to identify interactions between human and ZIKV proteins. Despite this methodological difference compared to our *in silico* approach, we could observe some overlaps about the AKT-mTOR signaling pathway, cell-cycle regulation pathway, and more specifically the modulation of ATM/ATR proteins in this pathway associated with the DNA-damage pathway. Other overlaps can be found when compared to our study with others that use the methodology *in silico*. *Ganapathiraju, Karunakaran & Correa-Menéndez (2016)* found pathways such as toll-like receptor signaling, axonal guidance signaling, and actin cytoskeleton signaling, all found in our results, while Esteves et al. (*Esteves et al., 2017*) found numbers of PPI predictions comparable with our numbers. Even in Lasso et al. (*Lasso et al., 2019*), where many viruses including ZIKV were evaluated, apoptosis, cell-cycle, JAK-STAT, and NF-κB pathways were "recurrently targeted across all viruses" and overlaps with our results. Those facts can indicate the robustness of the approach used here to predict protein interactions.

## CONCLUSIONS

Since the success of the machine learning approach depends on the set used to train the model, the present study deployed, with some improvements, a methodology described in the literature with a positive sample set based on experimental data. The negative samples were derived from the positive data using an approach focusing on the probability that a pair is non-interactive. The consensus network of PPI predictions made by Random Forest and SVM algorithms allowed an enrichment analysis that corroborates many aspects found in literature about ZIKV infection. The pathways presented on enrichment analysis are mainly related to viral infection, neuronal development, and immune response, all compatible with ZIKV infection, and there were different pathways among the two ZIKV strains compared. The strain PE243 showed more predicted interactions between proteins from the JAK-STAT signaling pathway when compared with the strain FSS13025, being this pathway specifically enriched for strain PE243, which could lead to a more disturbed immune response to this strain. In addition, some pathways were supported by experimental evidence in previous work of our group. These results show that the methodology employed here is solid, can reveal potential new interactions between the ZIKV and human cells, and can be used in other viral-host protein interaction studies.

## ACKNOWLEDGEMENTS

The authors thank FIOCRUZ-PE for the use of its facilities and the Bioinformatics research group for all the support.

### Funding

This work was supported by Fundação Oswaldo Cruz (FIOCRUZ). The computational platform used to apply the methodology described in this study was funded by Coordenação de Aperfeiçoamento de Pessoal de Nível Superior (CAPES, Brazil). The funders had no role in study design, data collection and analysis, decision to publish, or preparation of the manuscript.

### Grant Disclosures

The following grant information was disclosed by the authors:
Fundação Oswaldo Cruz (FIOCRUZ).
Coordenação de Aperfeiçoamento de Pessoal de Nível Superior (CAPES).

### Competing Interests

The authors declare that they have no competing interests.

### Author Contributions

- João Luiz de Lemos Padilha Pitta conceived and designed the experiments, performed the experiments, analyzed the data, prepared figures and/or tables, authored or reviewed drafts of the paper, and approved the final draft.
- Crhisllane Rafaele dos Santos Vasconcelos conceived and designed the experiments, performed the experiments, analyzed the data, prepared figures and/or tables, and approved the final draft.
- Gabriel da Luz Wallau conceived and designed the experiments, authored or reviewed drafts of the paper, and approved the final draft.
- Túlio de Lima Campos conceived and designed the experiments, authored or reviewed drafts of the paper, and approved the final draft.
- Antonio Mauro Rezende conceived and designed the experiments, analyzed the data, prepared figures and/or tables, authored or reviewed drafts of the paper, and approved the final draft.

### Data Availability

The raw data are available in the Supplemental Files.

### Supplemental Information

Supplemental information for this article can be found online at http://dx.doi.org/10.7717/peerj.11770#supplemental-information.

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
