# Peer review of "In silico predictions of protein interactions between Zika virus and human host"

_PeerJ, doi:10.7717/peerj.11770_

## Round 0.1 · original submission · Major Revisions

PeerJ thanks you for submitting this manuscript for review. I believe that your methodology and findings could merit publication after substantial changes to address the primary concerns that were raised the by the reviewers. I understand that there are conflicting opinions voiced by the reviewers; however, I would hope that you would look into the primary underlying concerns of:

1) citing additional introductory literature

2) justifying why the authors selected the dataset, and explain why any other ZIKV-related datasets were excluded from the analysis

3) Improving the explanation and interpretation of the enrichment results

4) Editing for grammar

Reviewer 1 ·

Basic reporting

The literature review is not sufficient. Only clinical background on Zika virus is presented. No review of the vast literature on protein protein interaction prediction, including prediction of Zika-human protein protein interactions is provided.

The images provided are of poor quality. Figure 3 is a "hairball" with little to no information provided by this image.

Figure 4 is an automatically retrieved image, which at best could be used in the supplement.

Experimental design

Computational methodology relies on adequate reference to prior work. If the same task has been carried out before, the authors must recreate the the prior work as to allow the side by side comparison of the new results. Otherwise, there is no way of evaluating if the current work constitutes an improvement over prior existing work.

The research question is not well defined. We are many years past predicting host-pathogen interactions if there are no major conclusions derived from the predictions or the methodology shows any novelty.

In its current form, it is impossible to conclude if the current work fills an actual knowledge gap.

Validity of the findings

The choice of the negative dataset is not well justified and researched. There are other approaches that previous PPI prediction work has considered and these should at least be referenced, with a justification for why the chosen approach is superior. Currently, it appears arbitrary why the Eid et al paper was chosen.

The Go-term enrichment yielded very generic pathways, such as ErbBx signaling, complement, NFkappa B, PI3K-AKt signaling and phosphatidylinostiol signaling, ubiquitin etc. These are not specific and it is not clear what we learn from finding the respective proteins over-represented in the interactome.

There is essentially no conclusion in this paper.

Additional comments

Here is an experimental determination of PPI large scale involving Zika - this is the minimum paper I would expect to see cited:

Comparative Flavivirus-Host Protein Interaction Mapping Reveals Mechanisms of Dengue and Zika Virus Pathogenesis.
Shah PS, Link N, Jang GM, Sharp PP, Zhu T, Swaney DL, Johnson JR, Von Dollen J, Ramage HR, Satkamp L, Newton B, Hüttenhain R, Petit MJ, Baum T, Everitt A, Laufman O, Tassetto M, Shales M, Stevenson E, Iglesias GN, Shokat L, Tripathi S, Balasubramaniam V, Webb LG, Aguirre S, Willsey AJ, Garcia-Sastre A, Pollard KS, Cherry S, Gamarnik AV, Marazzi I, Taunton J, Fernandez-Sesma A, Bellen HJ, Andino R, Krogan NJ.
Cell. 2018 Dec 13;175(7):1931-1945.e18. doi: 10.1016/j.cell.2018.11.028.
PMID: 30550790

There are many many other papers that are relevant that you should look at, some example are listed below:

https://www.ncbi.nlm.nih.gov/pmc/articles/PMC5747333/


https://www.ncbi.nlm.nih.gov/pmc/articles/PMC5742907/

https://www.ncbi.nlm.nih.gov/pmc/articles/PMC7102568/

https://www.ncbi.nlm.nih.gov/pmc/articles/PMC6474419/

https://www.ncbi.nlm.nih.gov/pmc/articles/PMC6783930/

https://www.ncbi.nlm.nih.gov/pmc/articles/PMC6736651/

·

Basic reporting

The main idea of this article is to use machine learning algorithms in order to evaluate novel ZIKV-Human protein interactions based on two viral strains, one of them related to neurological development issues.
The manuscript is well balanced, well organized, and the references used are sufficient for this study. Additionally, the study was presented with good scientific rigor, as well as with the quality of its figures, tables, and raw data.
The achieved results are well related to the manuscript hypothesis and give us a novel ZIKV-Human interactions mechanism, as well as confirms other studies cited in the discussion.
On the other hand, it should have a general English language revision in different parts of the text for ensuring the most appropriate formal and scientific language, but nothing that compromises the quality of this work.
The authors even referred “Eid, ElHefnawi & Heath, 2015” as its main reference for the training method. Is there any other study that used the same approach? I think it is important to cite another reference to reinforce the use of this methodology. Additionally, please check the same reference in the Pubmed, but for 2016:

Eid FE, ElHefnawi M, Heath LS. DeNovo: virus-host sequence-based protein-protein interaction prediction. Bioinformatics. 2016 Apr 15;32(8):1144-50. doi: 10.1093/bioinformatics/btv737. Epub 2015 Dec 16. PMID: 26677965.

Experimental design

The study was defined with a rigorous experimental design, using previous studies methods and validated databases and tools, as well as commonly used machine learning algorithms for evaluating protein interactions. On the other hand, it is not clear why the authors choose just that specific strain for comparison and did not include other ZIKV strain data for comparison. In this case, the only way to do this comparison is by using published results. I suggest, if it is possible that the authors include in the training set at least one more ZIKV strain for positive interactions. Additionally, please describe explaining the reasons you choose these specific strains.

Validity of the findings

Even other authors have used similar approaches to find the ZIKV-Human PPI network, this manuscript brings novel information about the possible mechanisms of this viral infection mainly in cases with neurological issues. The authors included additional training and validation steps, in comparison to other studies, and they found possible new mechanisms of interaction. Additionally, the authors confirmed other studies' findings which were cited in the discussion section. The results and discussion of this manuscript are well described and it is related to its main hypothesis, as well as is in the same way as other previous studies described for ZIKV-Human interactions and for other viruses.

---

## Round 0.2 · Minor Revisions

We thank you for resubmitting your updated manuscript and recognize the updates that were made. After an additional round of review overall there is good justification for requesting minor revisions to your manuscript before moving forward--especially those raised by reviewer #3.

Reviewer 1 ·

Basic reporting

The edits made in the revision introduced multiple grammatical and typographical errors.
Literature references were enhanced and additional references were included but their critical evaluation can be improved instead of "listing".
The "hairball" figure remains unacceptable in presentation and information content.

Experimental design

Thank you for the clarification on Kegg vs GO enrichment. Regardless of the source of enrichment, my concern remains that generic pathways were identified.
The research question is broad but valid, but the conclusions lack specificity.

Validity of the findings

Conclusions need to be specific to Zika and explained in the context of Zika biology.

Additional comments

"respectful disagreement" is not sufficient to address my concerns

·

Basic reporting

The authors have fixed all issues and followed all recommendations in this second version of the manuscript. Thus, I have no comments in this section.

Experimental design

The authors have fixed all issues and followed all recommendations in this second version of the manuscript. Thus, I have no comments in this section.

Validity of the findings

The authors have fixed all issues and followed all recommendations in this second version of the manuscript. Thus, I have no comments in this section.

Additional comments

Thank you for accepting my suggestions. I understand that the manuscript is now suitable for publication.

Reviewer 3 ·

Basic reporting

In this work, Pitta and colleagues applied Random Forest and SVM algorithms to predict the interaction between Zika virus and human proteins. The article is of broad interest and can help in finding targets for the development of new drugs. However, there are a few points that should be addressed before its ready for publication. Of note, the presentation of the data could be improved.

Experimental design

no comment

Validity of the findings

no comment.

Additional comments

Minor
The English language should be improved to ensure that an international audience can clearly understand your text. This includes some mistyping errors, such as abstract "on the other hands"->"on the other hand"; Line 232 Virus Mentha, but LIne 242 -> VirusMentha, what is the right way? Line 335 "non-interaction: ...

Important
The authors should use a figure to show the data in Table 2. There are many types of visualization for enrichment data, such as observed here: (https://bioconductor.statistik.tu-dortmund.de/packages/3.8/bioc/vignettes/enrichplot/inst/doc/enrichplot.html - Dot Plot).

Check figure quality. Figures 2, 3 and 4 seem below acceptable quality. However, it could be a product of PDF conversion. Consider checking prior publication.

Why the author did not account differentially structural and non-structural proteins, since structural proteins are more likely to be exposed to host proteins due to their role in particle formation?

---

## Round 0.3 · Minor Revisions

I appreciate your willingness to submit a revised version of your manuscript. My opinion is that this manuscript needs only a small number of relatively minor revisions to satisfy Reviewer 4. I would encourage you to especially focus on the reviewer feedback for the figures and tables.

Reviewer 4 ·

Basic reporting

This reviewer has been invited to comment on the manuscript starting from the rebuttal. Thus, my comments and arguments were built after suggestions from other reviewers. In general lines, the manuscript is sound and fit for publication after minor improvements described below.

Important: Authors should include a supplementary table with access ID for each protein shown in figure 2, divided by ZIKV-protein interactant. It must contain columns listing all important parameters obtained from PPI predictions. This will allow readers to get in touch with the data and further validate findings, thus improving visibility of this work. Moreover, will also allow the investigation of interactions with borderline significance as exemplified in lines 405-413.

I agree with other reviewers that Figure 4 is pointless. No information is added other than a ‘beautiful plot’ with different numbers of circles. The information contained in this figure should be better shown as a simple Venn diagram (or Euler diagram) associated with figure 2, where these numbers stand out without being rationally compared. Then, rephrase lines 345-348 and move where figure 2 is described. Also, a Venn diagram would allow quantification when this data is discussed in lines 420-425.

Experimental design

Experimental design seems suitable for their findings and is well documented.

Validity of the findings

Despite no validations, authors make it clear that their work is based on predictions and should be validated prior any assumptions.

Additional comments

This work is important as it highlights mechanistic differences between African and Asian strains of Zika virus. An example of these differences has been recently published by Aubry et al, 2021 Nat Comm (DOI 10.1038/s41467-021-21199-z), however Aubry et al do not propose any mechanism for their observations. This work by Pitta et al is important as it shed light on the differences between host-pathogen interaction of ZIKV strains. Moreover, their methodology could be easily implemented to investigate PPI between flaviviruses of medical importance and the human host.

---

## Round 0.4 · accepted · Accept

Thank you for your continued effort to address the reviewers' comments. I believe that these adjustments substantially improved the manuscript.